# ERDE: Entropy-Regularized Distillation for Early-exit

## Abstract

Although deep neural networks and in particular Convolutional Neural Networks have demonstrated state-of-the-art performance in image classification with relatively high efficiency, they still exhibit high computational costs, often rendering them impractical for real-time and edge applications. Therefore, a multitude of compression techniques have been developed to reduce these costs while maintaining accuracy. In addition, dynamic architectures have been introduced to modulate the level of compression at execution time, which is a desirable property in many resource-limited application scenarios. The proposed method effectively integrates two well-established optimization techniques: early exits and knowledge distillation, where a reduced student early-exit model is trained from a more complex teacher early-exit model. The primary contribution of this research lies in the approach for training the student early-exit model. In comparison to the conventional Knowledge Distillation loss, our approach incorporates a new entropy-based loss for images where the teacher's classification was incorrect. The proposed method optimizes the trade-off between accuracy and efficiency, thereby achieving significant reductions in computational complexity without compromising classification performance. The validity of this approach is substantiated by experimental results on image classification datasets CIFAR10, CIFAR100 and SVHN, which further opens new research perspectives for Knowledge Distillation in other contexts.

## 1 Introduction

In the field of deep learning, the reduction of computational cost is a matter of significant concern. A multitude of techniques have been identified that adapt Convolutional Neural Network (CNN) architectures, thereby reducing both computational cost and model size. These techniques have proven to be efficient in various aspects of architecture optimization, where the combination of methods is a particularly effective approach to achieving high compression rates when they are complementary, such as a pruning and a quantization method, for example.

For instance, Qi et al. (2021) achieve a compression rate of 50% with an accuracy reduction of only 0.15%-0.37%. Another method proposed a combination early-exit and quantization on CNN (Li et al., 2022) reaching a compression reduction of 50% with an accuracy drop of 1% to 3%. Quantization can also be effectively combined with pruning, e.g. Song Han (2016). Thus, the combination of distinct methods allows to further optimize the trade-off between compression and accuracy.

In this study, we integrate the knowledge distillation and early-exit approaches to obtain distilled dynamic neural networks. That is, at inference time, the distilled network can further reduce its compression rate dynamically by executing only parts of the model depending on external conditions (e.g. battery power) or internal criteria (e.g. the difficulty of the input).

Knowledge distillation (KD) is a training method in which a large reference network, designated as the "teacher", is employed to train a smaller network, referred to as the "student". This approach enables the student network to attain a higher accuracy compared to training without KD. Early-exit (EE), also called multi-exit, methods employ a dynamic compression technique where the network makes multiple intermediate predictions after executing a certain number of layers. The complexity of the model can thus be modulated (automatically or manually) at run-time by exiting earlier from the neural network and thereby eliminating the need for subsequent calculations.

The proposed method named ERDE for Entropy-Regularized Distillation for Early-exit relies on a specific training method that allows us to apply knowledge distillation between two early-exit networks. By effectively combining the two approaches, we are able to cumulate the compression gains of both and, in addition, obtain a run-time configurable model.

In summary, our contributions are the following:

- We present a new compression approach that effectively combines Knowledge Distillation with Early-Exit dynamic neural network architectures.

- We introduce a new distillation method based on a loss function applied at intermediate exits trying to maximize the entropy for those examples where the teacher model is uncertain.

We have tested our approach with different CNNs on three classical image classification benchmarks. Our models obtained a significant reduction in computational complexity, i.e. up to around 10 times lighter than the original models, with little loss in accuracy or even a gain in some cases. In addition, compared to the standard KD training algorithm, applying KD to EE architectures with our proposed entropy-regularization loss improves the average accuracy on all tested datasets for all possible EE thresholds.

## 2 Related Work

A broad spectrum of model compression strategies has been proposed to alleviate the memory footprint and computational complexity of deep neural networks. These approaches primarily encompass parameter quantization, which reduces numerical precision; network pruning, which removes redundant weights, neurons, or structural components; and Neural Architecture Search, which automates the discovery of compact and efficient model topologies. Collectively, these techniques aim to achieve favorable accuracy–efficiency trade-offs while preserving predictive performance.

### 2.1 Dynamic Neural Networks

The need for dynamic networks comes from the inability of static ones to adapt the computational graphs or the network parameters. For example, processing complex images requires deep and complex networks that require more computation. However, this type of network will do a large amount of unnecessary computation when processing simpler images.

Many techniques have been developed to adapt the architecture of neural networks at execution time (Han et al., 2021). We can divide these methods into three categories: dynamic depth, dynamic width, and dynamic routing. Each category in turn contains several techniques.

Dynamic depth methods are designed to avoid the redundant computations of deep networks mentioned above. This can be achieved by exiting at shallow exits for simple inputs (early exit) as in Rahmath et al. (2024) or by selectively skipping certain intermediate layers given a particular input (layer skipping) as in Graves (2016) and Wang et al. (2017). Dynamic width methods are more fine-grained than the previous ones, all layers are executed but some units (neurons, channels or branches) are not activated depending on the input. One of the dynamic width methods is Mixture of Experts (MoE) (Jacobs et al., 1991; Eigen et al., 2013). It is a strategy that dynamically selects and uses only a small part of the network (the experts) for processing, based on the given input. Dynamic routing is a key mechanism in modern dynamic neural networks that adapt their computational pathways based on input. It enables the network to optimize its structure by selectively activating specific layers, channels, or paths within architectures like SuperNets (Cha et al., 2022). We can consider early-exiting networks as a special form of SuperNets. Another type of dynamic routing networks are CapsuleNets, where routing between capsules captures hierarchical relationships (Sabour et al., 2017; Hinton et al., 2018).

## 2.2 Early-Exit Approaches

Early-exit methods add various exits at different layers and are composed of additional classification branches (denoted "exit branches" in the following). The inference is initially performed up to the first exit branch. At this point, a prediction is made with an associated confidence. If the confidence reaches a certain threshold, the inference is stopped and the resulting prediction is returned, otherwise the inference continues until the next exit.

The exit branches can be composed of one or more layersdiffer that can be of different types: either a classic single Fully Connected (FC) layer (Chen et al., 2020; Dai et al., 2020), multiple FC layers (Zhao et al., 2021) or more complex architectures adding one or more convolution layers and pooling layers before the FC layer (Teerapittayanon et al., 2016; Jo et al., 2023). Note that the calculations performed on these exit branches are "lost" if the confidence is below the threshold and thus the exit not used, so it is important to ensure that the exits remain cost effective. The placement of the exit branches may also differ. Some methods add an exit after each layer (Dai et al., 2020), others choose specific layers (Berestizshevsky & Even, 2018; Teerapittayanon et al., 2016) and finally there are more complex techniques like using metrics or gating function to decide where to put the exits (Fang et al., 2020; Li et al., 2023).

The inclusion of additional exits to a classical Deep Neural Network (DNN) entails a different training strategy because the network has several outputs that need to be optimized. Joint training is the most trivial method, in which all branches are trained simultaneously (Teerapittayanon et al., 2016), where the global loss is defined as the sum of all losses obtained at the end of each branch considered by a chosen coefficient. In the branch-wise training strategy, each side branch is trained separately, together with the preceding layers of the backbone DNN (Belilovsky et al., 2019). Separate training consists of treating the exit branches as independent classifiers and training them independently (Chiu et al., 2023). Another training strategy, called two-stage training, is to train the DNN backbone first, then its parameters are frozen and the exit branches are trained separately (Berestizshevsky & Even, 2018). Knowledge Distillation (KD) based training uses the exit branches as student models that learn from the output of the DNN backbone (He et al., 2015; Hinton et al., 2014).

Furthermore, many exit policies have been developed to decide whether to exit or not depending on the branch output. These policies can be divided into two categories: static (rule-based) policies and dynamic (learnable) policies. Static exit policies measure the confidence in the predictions made using metrics such as entropy, maximum softmax or user-defined scoring functions. A single threshold can be applied to all branches, or the threshold can be specified for each branch. These methods are easy to implement and fast, but lack robustness due to their inability to adapt. On the other hand, some learnable exit policies have been proposed, such as exit selection controllers (Dai et al., 2020; Guan et al., 2017), reinforcement learning (Guan et al., 2017; Wang et al., 2017), and soft gating mechanisms (Mullapudi et al., 2018). They have a high training complexity but also a high robustness to diverse input.

## 2.3 Distillation and Dynamic Neural Networks

One of the commonly used compression methods is called Knowledge Distillation (Hinton et al., 2014). It consists in using a reference model (teacher) to train a smaller network (student), leading to a better accuracy in a shorter time compared to the same student trained without KD. This type of optimisation has been applied to many other techniques to obtain better results. For example, when pruning a network, some work like Chen et al. (2021) shows that using the unpruned network as a teacher helps to obtain better accuracy than classical retraining.

Inspired by Knowledge Distillation, Zhang et al. (2019) have developed self-distillation, which unlike traditional KD, works within a single network. It is a training technique to improve model performance, not a method to compress models. The idea of this method is to divide the network into different sections and distill the knowledge from the deeper sections to the shallower ones. This approach not only improves accuracy but also promotes computational efficiency.

Building on the principles of self-distillation, other researchers have explored its potential in terms of optimisation. For example, ESCEPE (Gourtani & Meratnia, 2023) is an approach based on weight clustering

and self-distillation achieving a high compression ratio of the early-exit network with minimal impact on the accuracy of the intermediate classifiers. The proposed method combines early-exit with self-distillation: a special case of knowledge distillation where the teacher is the network before compression. As apposed to our approach, this work does not explore the possibility of using a different network as a teacher and focuses mainly on pruning a network.

Runtime Neural Pruning (RNP) (Lin et al., 2017) is a framework that prunes networks according to the input and feature maps. Unlike traditional methods that create a fixed pruned model for deployment, RNP preserves the full capacity of the original network, allowing for adaptable pruning. The pruning process is modelled as a Markov Decision Process, and a reinforcement learning agent is trained to determine the optimal pruning policy. Unlike our method, RNP does not explore knowledge distillation at all.

Finally, Xu et al. (2022) propose a multiple-stage knowledge distillation method. It involves the construction of intermediate exits on top of classic networks, facilitating the distillation of knowledge through various points of the network without utilizing the created "early-exit" for dynamic purposes. Additionally, the proposed One-to-One MSKD model exclusively computes the cross-entropy classification loss and the Kullback–Leibler (KL) distillation loss. ERDE proffers the utilization of intermediate exits as a strategy to curtail computations and employ an entropy loss on erroneously classified samples to augment network confidence.

A comparison will be made between the proposed architecture and MSDNet (Huang et al., 2017), a multi-scale densely connected architecture designed for anytime prediction. The model maintains parallel feature representations at multiple resolutions and incorporates intermediate classifiers to enable adaptive computation with favorable accuracy–efficiency trade-offs.

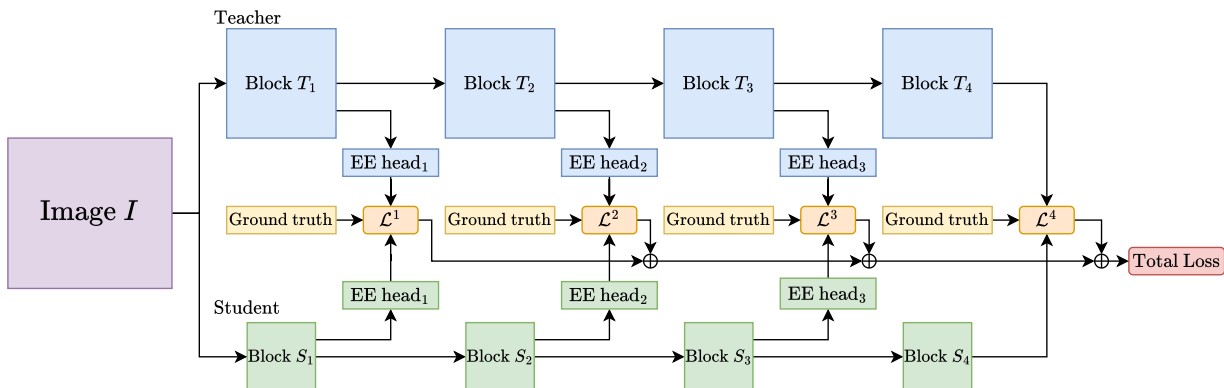

Figure 1: Our ERDE architecture and training approach for 4 teacher and student blocks (in blue and green respectively). The $\mathcal{L}^i$ correspond to the different loss functions ($\mathcal{L}_{CE}$, $\mathcal{L}_{KD}$, and $\mathcal{L}_{E}$) at the $i$-th exit. At inference, only the compressed student model is used.

## 3 Method

This section describes the methodology utilized and the contribution of the present study. The proposed architecture (illustrated in Figure 1) uses early exits with $n-1$ branches, i.e. there are $n-1$ early exit heads, and the $n$-th one is the classical exit of the network. The architecture of the EE heads will be defined in the subsequent section. As previously outlined, the premise of early exits is that the example progresses through the network until the first exit. At this point, an initial prediction is generated, accompanied by the attribution of an uncertainty score $c_i$ to the prediction. This score is defined as the entropy of the softmax vector of the prediction. If the uncertainty score is below a predetermined threshold, the prediction is designated as the network's output, i.e., the final prediction. This predetermined threshold, denoted as $\theta$ in $[0; \infty[$, will be consistent for all exits. As the threshold increases, more predictions satisfy the early-exit criterion, resulting in a greater proportion of samples exiting at earlier branches. Conversely, decreasing the threshold imposes a stricter confidence requirement, causing more samples to propagate to deeper exits.

If the score remains above the threshold, the image $I$ continues through the network until the next exit, where the same procedure is repeated. If the network does not hold sufficient confidence in a prediction at the $n-1$-th exit, the image will terminate at the ultimate exit (corresponding to the conventional end of the network). It can thus be concluded that utilising a threshold of $\theta = 0$ is analogous to deactivating the early-exit. In this study, entropy was selected as the uncertainty metric, a common choice in the literature (Teerapittayanon et al., 2016)).

We did not explore further EE architectures or policies because we believe this is not part of our contribution. Algorithm 1 formalizes the early-exit inference mechanism.

---

**Algorithm 1** Early-exit inference algorithm.

$i \leftarrow 0$
$h_0 \leftarrow I$
$c_0 \leftarrow \infty$
**while** $c_i > \theta$ and $i \leq n$ **do**
    $i++$
    $h_i \leftarrow \text{Block}_i(h_{i-1})$
    $y_i \leftarrow \text{EE head}_i(h_i)$
    $c_i \leftarrow \text{entropy}(\text{softmax}(y_i))$
**end while**
**return** $y_i$

---

The training of a multi-exit network necessitates a modification of the conventional training process. This modification is required to ensure the training of all heads and the calculation of the confidence threshold, which is a prerequisite in certain cases. The objective is to consider the various exit losses. The subsequent paragraph provides a detailed definition of these losses.

Firstly, the following notation must be established: $y_{T_i} = \text{Softmax}(z_{T_i}, T)$, the softmax of the $i$-th exit of the teacher; $y_{S_i} = \text{Softmax}(z_{S_i}, T)$, the softmax of the $i$-th exit of the student. Where $T$ is the temperature parameter of the softmax function. Assume, that the number of classes of our problem is $K$ and the number of branches of both, teacher and student, is $n$.

The Cross-Entropy (CE) loss is one of the classic loss used in deep learning. It is defined as the cross entropy between the ground truth and the student prediction

$$\mathcal{L}_{\text{CE}}^i = - \sum_{k \in K} y_k \log\left(\hat{y}_{S_{i_k}}\right) \ . \tag{1}$$

This loss compared the class given by the student with the ground truth. $\mathcal{L}_{\text{CE}}^i$ is also used to train the teacher network in the following experiments.

In KD, the classic loss employed is defined as the sum of two losses: the KD loss and the CE loss. The KD loss is expressed as the difference between the student's output and the teacher's output. This loss is designed to encourage the student to make a prediction that mirrors the teacher's prediction. It can be defined through various functions; however, the Kullback-Leibler divergence, which was utilized in the original work of Hinton et al. (2014), was selected for implementation in this study:

$$\mathcal{L}_{\text{KL}}^i = \frac{T^2}{K} \times \sum_{k \in K} \hat{y}_{T_{i_k}} \log\left(\frac{\hat{y}_{T_{i_k}}}{\hat{y}_{S_{i_k}}}\right) \ . \tag{2}$$

In order to train the student and take into account the early exit, we adapted the classic knowledge distillation method. The primary contribution of the proposed method is the division of the loss between two cases: one for samples that are correctly classified by the teacher and one for samples that are incorrectly classified. In the event that the teacher has successfully classified the sample, we use the combination of $\mathcal{L}_{\text{KL}}$ and $\mathcal{L}_{\text{CE}}$ as

usual. In the alternative scenario, the entropy of the output vector is subtracted. Specifically, this entails the subtraction of a quantity $\mathcal{L}_E$ represented as:

$$\mathcal{L}_E^i = - \sum_{k \in K} \hat{y}_{S_{i_k}} \log \left( \hat{y}_{S_{i_k}} \right) \ .$$ (3)

The idea of this entropy loss is to force the network to be "uncertain" for samples where the teacher failed to accurately predict the class of the image. The maximum of this loss for a sample is achieved when all classes are equally predicted and is minimized when a class is predicted with a probability of 1.

Finally the overall loss can be written as:

$$\mathcal{L}_{tot} = \sum_{i=0}^{n-1} \mathbf{1}_{y=\hat{y}_{T_i}} (\omega_{KL} \mathcal{L}_{KL}^i + \omega_{CE} \mathcal{L}_{CE}^i) - \mathbf{1}_{y \neq \hat{y}_{T_i}} \omega_E \mathcal{L}_E^i$$
$$+ \omega_{KL} \mathcal{L}_{KL}^n + \omega_{CE} \mathcal{L}_{CE}^n \ .$$ (4)

At the final exit, we employ the conventional KD loss regardless of the accuracy of the teacher prediction. This choice stems from the principle that the final exit is required to produce an output, even for instances where the confidence threshold is not attained. In contrast, the purpose of our entropy loss is to accentuate the uncertainty in intermediate exits, thereby preventing erroneous intermediate predictions.

## 4  Experiments

We evaluated our approach on three different standard datasets for image classification: CIFAR10, CIFAR100[1] and SVHN[2]. For SVHN, we did not use the optional extra training data in our experiments. We tested different student-teacher couples on these datasets including ResNet (He et al., 2016) and ConvNeXt architectures (Liu et al., 2022). We have used 3 different ResNet networks 34, 18, 10, and 8. Where ResNet8 is similar to a ResNet10 but with only 3 blocks (instead of 4 in a ResNet10). We added an exit branch after each block, where the last exit corresponds to the final output of the full model. To minimise the computational overhead, our exit branches are very shallow and operate on the last convolution layer of the preceding block. Early-exit head contain only a batch normalization layer, a ReLU activation function, a 2x2 average pooling layer, a Dropout with probability 0.5 and a single FC layer that performs the intermediate and final predictions.

Table 1: Performance comparison on CIFAR10 for different teacher–student combinations.

| Approach | Teacher | Student | Accuracy | MACs (M) | rel. MACs (%) | Latency (ms) |
|---|---|---|---|---|---|---|
| teacher | ResNet34 | | $94.44 \pm 0.23$ | 1160 | 100 | 7.4 |
| student w/o KD | | ResNet10 | $91.69 \pm 0.34$ | 214 | 21.9 | 4.38 |
| student KD | ResNet34 | ResNet10 | $\mathbf{93.27 \pm 0.21}$ | 214 | 21.9 | 4.21 |
| ERDE ($\theta = 0$) | ResNet34 | ResNet10 | $\mathbf{93.30 \pm 0.20}$ | 214 | 21.9 | 4.78 |
| ERDE ($\theta = 0.1$) | ResNet34 | ResNet10 | $92.42 \pm 0.14$ | **115** | **9.9** | **1.49** |
| teacher | ConvNeXt-b | | $93.03 \pm 0.19$ | 313 | 100 | 12.68 |
| student w/o KD | | ConvNeXt-t | $91.61 \pm 0.39$ | 91 | 29.9 | 8.09 |
| student KD | ConvNeXt-b | ConvNeXt-t | $93.27 \pm 0.17$ | 91 | 29.9 | 9.33 |
| ERDE ($\theta = 0$) | ConvNeXt-b | ConvNeXt-t | $\mathbf{93.33 \pm 0.06}$ | 91 | 29.9 | 9.26 |
| ERDE ($\theta = 0.2$) | ConvNeXt-b | ConvNeXt-t | $93.10 \pm 0.07$ | **34** | **10.8** | **4.43** |

For each of the datasets, we compared our model trained with our proposed distillation loss (4) to the following baselines: a teacher model without early exits, a student without KD and a student trained with

---

[1]https://www.cs.toronto.edu/ kriz/cifar.html

[2]http://ufldl.stanford.edu/housenumbers

classical KD using the sum of the cross-entropy loss and the distillation loss (KL divergence). In order to ensure the reliability of the results, it was necessary to undertake each experiment on five separate iterations, so that a mean result and its associated standard deviation could be obtained.

Table 2: Performance comparison on CIFAR100 for different teacher–student combinations.

| Approach | Teacher | Student | Accuracy ($\pm$ std) | MACs (M) | rel. MACs (%) | Latency (ms) |
|---|---|---|---|---|---|---|
| teacher | ResNet34 | | $74.47 \pm 0.68$ | 1163 | 100.0 | 7.89 |
| student w/o KD | | ResNet10 | $70.12 \pm 0.24$ | 257 | 22.1 | 4.10 |
| student KD | ResNet34 | ResNet10 | $73.74 \pm 0.32$ | 257 | 22.1 | 4.16 |
| ERDE ($\theta = 0$) | ResNet34 | ResNet10 | $\mathbf{73.93 \pm 0.25}$ | 257 | 22.1 | 4.04 |
| ERDE ($\theta = 0.6$) | ResNet34 | ResNet10 | $73.79 \pm 0.25$ | **145** | **12.5** | **3.35** |
| teacher | ConvNeXt-b | | $74.22 \pm 0.62$ | 314 | 100.0 | 12.81 |
| student w/o KD | | ConvNeXt-t | $71.28 \pm 0.77$ | 91 | 29.1 | 8.89 |
| student KD | ConvNeXt-b | ConvNeXt-t | $\mathbf{76.35 \pm 0.23}$ | 91 | 29.1 | 8.02 |
| ERDE ($\theta = 0$) | ConvNeXt-b | ConvNeXt-t | $\mathbf{76.36 \pm 0.20}$ | 91 | 29.1 | 9.07 |
| ERDE ($\theta = 1.0$) | ConvNeXt-b | ConvNeXt-t | $75.58 \pm 0.15$ | **49** | **15.5** | **5.64** |

We trained the early-exit models by simultaneously minimizing the loss for all exits, i.e. we simply minimized the sum of all losses without any weighting. The teacher and student models without KD are trained with the CE loss (1) without any (self-)distillation. The student models with classical KD are trained with

$$\mathcal{L}_{\mathrm{KD}} = \sum_{i=0}^{n} \omega_{\mathrm{KL}}\mathcal{L}_{\mathrm{KL}}^{i} + \omega_{\mathrm{CE}}\mathcal{L}_{\mathrm{CE}}^{i} \ , \tag{5}$$

and to evaluate our proposed approach we trained models with the loss in (4).

All models are trained for 300 epochs with a batch size of 64 and a learning rate of $10^{-3}$ with the Adam algorithm. To avoid overfitting, we applied early stopping with a validation set of 2000 images for CIFAR10 and CIFAR100 and 13256 images for SVHN. For the ResNet models (apart from the ResNet10) we used ImageNet-pretrained models. For the ConvNeXt models we started with ImageNet pre-trained weights, otherwise the models were not able to converge properly and did overfit too much. Furthermore, data augmentation is performed with random horizontal flips, rotations, translations, crops and random erasing. For all experiments, we set $\omega_{\mathrm{KL}} = 0.25$, $\omega_{\mathrm{CE}} = 0.75$, $\omega_E = 0.005$ and $T = 2$. The subsequent section will address the selection of the parameter $\omega_E$.

Table 3: Performance comparison on SVHN for different teacher–student combinations.

| Approach | Teacher | Student | Accuracy ($\pm$ std) | MACs (M) | rel. MACs (%) | Latency (ms) |
|---|---|---|---|---|---|---|
| teacher | ResNet34 | | $95.77 \pm 0.09$ | 1160 | 100.0 | 7.69 |
| student w/o KD | | ResNet10 | $94.81 \pm 0.27$ | 254 | 21.9 | 4.48 |
| student KD | ResNet34 | ResNet10 | $96.11 \pm 0.20$ | 254 | 21.9 | 4.12 |
| ERDE ($\theta = 0$) | ResNet34 | ResNet10 | $\mathbf{96.28 \pm 0.11}$ | 254 | 21.9 | 4.99 |
| ERDE ($\theta = 0.4$) | ResNet34 | ResNet10 | $95.77 \pm 0.10$ | **88** | **7.6** | **2.39** |
| teacher | ConvNeXt-b | | $94.86 \pm 0.21$ | 314 | 100.0 | 14.92 |
| student w/o KD | | ConvNeXt-t | $94.39 \pm 0.19$ | 91 | 29.0 | 8.03 |
| student KD | ConvNeXt-b | ConvNeXt-t | $95.37 \pm 0.12$ | 91 | 29.0 | 9.43 |
| ERDE ($\theta = 0$) | ConvNeXt-b | ConvNeXt-t | $\mathbf{95.59 \pm 0.06}$ | 91 | 29.0 | 8.20 |
| ERDE ($\theta = 0.6$) | ConvNeXt-b | ConvNeXt-t | $95.43 \pm 0.03$ | **24** | **7.7** | **3.73** |

For evaluating the different models, we computed the accuracy and the number of MACs (Multiply-Accumulate operations) and varied the confidence thresholds for the early-exit models. As the number of operations varies for each example, we report the average MACs (per example) over the whole test set. For each configuration, 5 training runs are performed with different random initialisations, to compute the mean and standard deviation of accuracies. The average latency is computed with a batch size of 1 on a NVIDIA V100 32 GB graphics card including all data loading and transfer overhead. We performed the latency computation on 180 images after a warm-up of 20 images.

In addition to the standard KD, a comparison is made between our method and MSDNet. Nevertheless, it is imperative to acknowledge that our proposed method should be regarded as a complementary compression method to the existing ones. To illustrate this point, consider the compression ratio in pruning approaches, which can typically be selected with a high degree of precision. In contrast, in KD, the target student architecture is predetermined.

## 5  Results

The overall performance results in terms of accuracy, MMACs and latency are shown in Table 1, 2, and 3. The teacher model demonstrates a higher degree of accuracy than that exhibited by the student model without knowledge distillation. However, when KD is employed, it is possible for the student to outperform the teacher (a result that is frequently observed in the context of KD).

It is noteworthy that the incorporation of our specific entropy loss during the training process has been shown to yield more favorable results in comparison to the conventional distillation loss regardless of the EE threshold and the tested teacher-student architectures. The discrepancy in accuracy between the classic KD and the proposed loss function can be marginal for specific configurations. However, our method consistently demonstrates superiority in terms of mean accuracy. In some cases, our approach increases the accuracy by more than 5 points comparing to the reference student (ConvNeXt-b/ConvNeXt-t on SVHN).

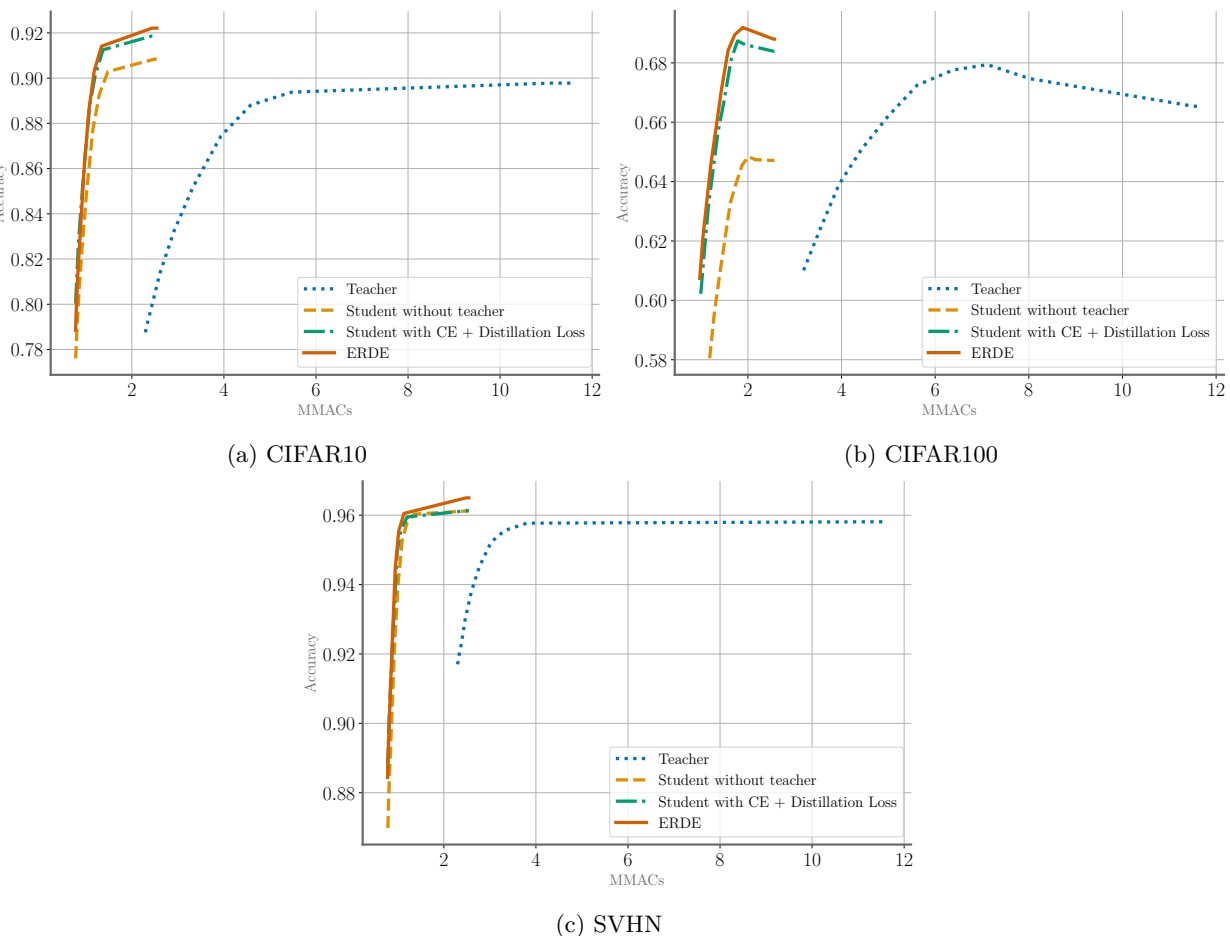

Figure 2: Test accuracy as a function of average MACs for different datasets and training strategies using ResNet34 as teacher and ResNet10 as student.

As illustrated in Figure 2a, 2b, and 2c, the early exit threshold has a significant impact on the performance of the models. It is noteworthy that decreasing this threshold can lead to a substantial reduction in the number of MACs without a concomitant decline in accuracy.

Surprisingly, for CIFAR100, it appears that the best accuracy is not obtained when the early exit threshold is maximized. This could be due to some overfitting occurring at the last exit of the model.

As shown in Table 1, 2, and 3, compared to the original teacher models, the models trained with our ERDE approach have only around 7%-15% of MAC operations depending on the chosen architectures with very little decrease in accuracy. In terms of latency, we can achieve a reduction of up to 4 times. In some cases, we may possibly go beyond this by choosing even smaller teacher models. For some results, we highlight the accuracy of two models when the difference in accuracy between them is too low to discriminate. However, even when the gap is narrow, ERDE outperforms classic KD systematically.

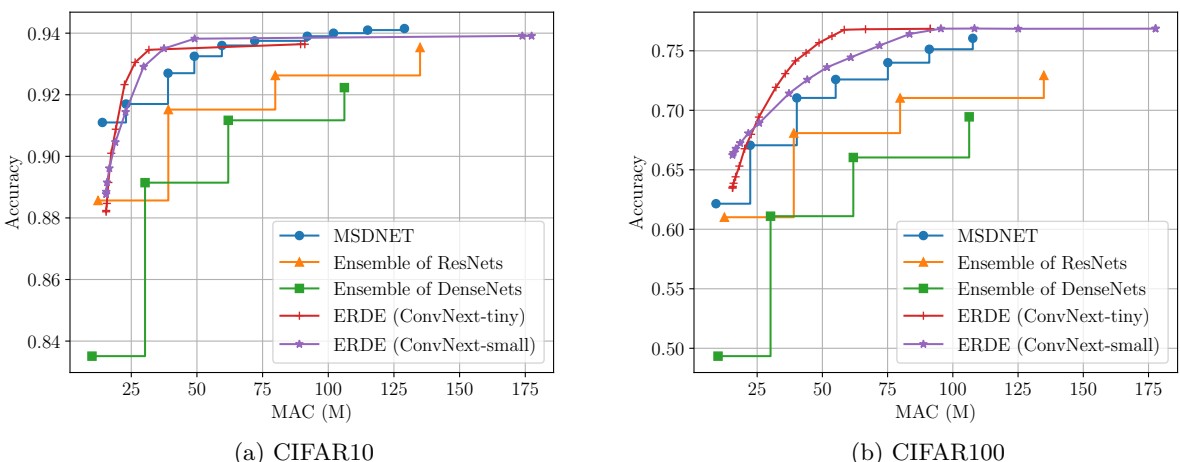

(a) CIFAR10                         (b) CIFAR100

Figure 3: Test accuracy as a function of average MACs for different datasets and methods.

A comparison was made between our model and MSDNet, a state-of-the-art network. A comparison with different depth ResNet and DenseNet networks is also conducted (see MSDNet (Huang et al., 2017) for more details). The results are displayed in Figure 3. On the CIFAR100 dataset (Figure 3b), the two ERDE networks (utilizing ConvNeXt-tiny and ConvNeXt-small as baselines) exhibited superior accuracy-to-MACs trade-off performance in comparison to MSDNet. In the case of CIFAR10 (Figure 3a), however, the outcomes are less obvious. The performance of our models demonstrates a clear superiority in the 15-65 MMAC range in comparison to MSDNET. However, MSDNET has been shown to exhibit a small advantage in terms of achieving optimal results outside this range.

We have studied the impact of $\omega_E$ on a ConvNeXt-tiny network trained using ConvNeXt-base as teacher on the three studied datasets. We have tested several values for $\omega_E$ from 0 to $5 \times 10^{-2}$ the resulting accuracy is shown in Figure 4. The best weight for the entropy loss is around $5 \times 10^{-3}$. The accuracy is rather stable in the range $[0; 0.02]$ but greater values lead to a significant decrease.

Table 4: Ablation study of the different loss configurations.

|  | All losses | KD only | ERDE |
|---|---|---|---|
| Accuracy ($\pm$ std) | $93.054 \pm 0.095$ | $93.265 \pm 0.170$ | $\mathbf{93.326 \pm 0.065}$ |

Finally, the effect of entropy loss applied to all samples was explored. The accuracy obtained by ConvNeXt-t models trained on CIFAR10 using ConvNeXt-b teacher models for different losses is shown in Table 4. It is interesting to note that, as hypothesised, the entropy loss, when applied on each occasion, had a penalising effect on the network rather than an improvement in its performance.

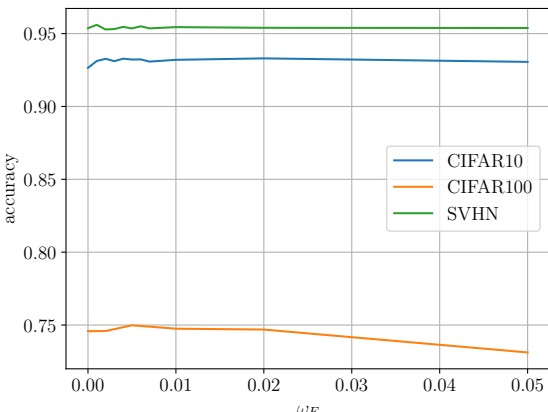

Figure 4: Accuracy of the student for different values of $\omega_E$

## 6 Conclusion

We have proposed an original method to apply knowledge distillation to an early-exit architecture. This is achieved by using a specific loss and training process. This approach effectively combines knowledge distillation with early exit architectures and thus leads to compact dynamic networks that can control the accuracy-complexity trade-off at run time and leverage both compression methods to reduce the complexity even further. The efficacy of our method has been demonstrated on three distinct datasets on different CNN architectures and different teacher-student combinations. Our method systematically outperforms conventional knowledge distillation and is able to reduce the computational complexity up to around 10 times with no (or negligible) decrease in accuracy. It also outperformed MSDNet, ResNet and DenseNet models.

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

## A    More tested couples

Results obtained on single experiments performed on non-pretrained networks.

Table 5: Performance comparison for CIFAR10 for different teacher-student combinations. Reported MMACs and latencies are averages over the whole test dataset.

| Approach | Teacher | Student | Accuracy | MACs (M) | rel. MACs (M) |
|---|---|---|---|---|---|
| teacher | ResNet18 | | 0.9153 | 556.4 | 100% |
| student w/o KD | | ResNet10 | 0.9114 | 254.2 | 45.7% |
| student w KD | ResNet18 | ResNet10 | 0.9275 | 254.2 | 45.7% |
| ERDE ($\theta = 0$) | ResNet18 | ResNet10 | **0.9300** | 254.2 | 45.7% |
| ERDE ($\theta = 0.4$) | ResNet18 | ResNet10 | 0.9082 | **116.8** | **21.0%** |
| teacher | ResNet34 | | 0.8978 | 1160.8 | 100% |
| student w/o KD | | ResNet8 | 0.9009 | 195.4 | 16.8% |
| student w KD | ResNet34 | ResNet8 | 0.9057 | 195.4 | 16.8% |
| ERDE ($\theta = 0$) | ResNet34 | ResNet8 | **0.9060** | 195.4 | 16.8% |
| ERDE ($\theta = 0.4$) | ResNet34 | ResNet8 | 0.8856 | **103.5** | **8.9%** |
| teacher | CNXT-b | | 0.9070 | 313.6 | 100% |
| student w/o KD | | ResNet8 | 0.9009 | 195.4 | 62.3% |
| student w KD | CNXT-b | ResNet8 | 0.9041 | 195.4 | 62.3% |
| ERDE ($\theta = 0$) | CNXT-b | ResNet8 | **0.9066** | 195.4 | 62.3% |
| ERDE ($\theta = 0.6$) | CNXT-b | ResNet8 | 0.8883 | **109.4** | **34.9%** |

Table 6: Performance comparison for CIFAR100 for different teacher-student combinations. Reported MMACs and latencies are averages over the whole test dataset.

| Approach | Teacher | Student | Accuracy | MACs (M) | rel. MACs (M) |
|---|---|---|---|---|---|
| teacher | ResNet18 | | 0.6758 | 559 | 100% |
| student w/o KD | | ResNet10 | 0.6521 | 256.8 | 45.9% |
| student w KD | ResNet18 | ResNet10 | 0.6702 | 256.8 | 45.9% |
| ERDE ($\theta = 0$) | ResNet18 | ResNet10 | **0.6741** | 256.8 | 45.9% |
| ERDE ($\theta = 0.4$) | ResNet18 | ResNet10 | **0.6726** | **172.6** | **30.9%** |
| teacher | ResNet34 | | 0.6654 | 1163.4 | 100% |
| student w/o KD | | ResNet8 | 0.6462 | 198 | 17.0% |
| student w KD | ResNet34 | ResNet8 | 0.6504 | 198 | 17.0% |
| ERDE ($\theta = 0$) | ResNet34 | ResNet8 | **0.6578** | 198 | 17.0% |
| ERDE ($\theta = 0.6$) | ResNet34 | ResNet8 | **0.6536** | **133** | **11.4%** |
| teacher | CNXT-b | | 0.7029 | 314 | 100% |
| student w/o KD | | ResNet8 | 0.6462 | 198 | 63.1% |
| student w KD | CNXT-b | ResNet8 | 0.6427 | 198 | 63.1% |
| ERDE ($\theta = 0$) | CNXT-b | ResNet8 | **0.6530** | 198 | 63.1% |
| ERDE ($\theta = 1.4$) | CNXT-b | ResNet8 | 0.6374 | **133** | **42.4%** |

Table 7: Performance comparison for SVHN for different teacher-student combinations. Reported MMACs and latencies are averages over the whole test dataset.

| Approach | Teacher | Student | Accuracy | MACs (M) | relative MACs |
|---|---|---|---|---|---|
| teacher | ResNet18 | | 0.9623 | 556.4 | 100% |
| student w/o KD | | ResNet10 | 0.9592 | 254.2 | 45.7% |
| student w KD | ResNet18 | ResNet10 | 0.9637 | 254.2 | 45.7% |
| ERDE ($\theta = 0$) | ResNet18 | ResNet10 | **0.9657** | 254.2 | 45.7% |
| ERDE ($\theta = 0.4$) | ResNet18 | ResNet10 | 0.9624 | **117.3** | **21.1%** |
| teacher | ResNet34 | | 0.9584 | 1160.8 | 100% |
| student w/o KD | | ResNet8 | 0.9572 | 195.4 | 16.8% |
| student w KD | ResNet34 | ResNet8 | 0.9599 | 195.4 | 16.8% |
| ERDE ($\theta = 0$) | ResNet34 | ResNet8 | **0.9607** | 195.4 | 16.8% |
| ERDE ($\theta = 0.2$) | ResNet34 | ResNet8 | 0.9566 | **105.3** | **9.1%** |
| teacher | CNXT-b | | 0.9476 | 313.6 | 100% |
| student w/o KD | | ResNet8 | 0.9572 | 195.4 | 62.3% |
| student w KD | CNXT-b | ResNet8 | 0.9549 | 195.4 | 62.3% |
| ERDE ($\theta = 0$) | CNXT-b | ResNet8 | **0.9574** | 195.4 | 62.3% |
| ERDE ($\theta = 1.6$) | CNXT-b | ResNet8 | **0.9557** | **110.1** | **35.1%** |

