# OpenReview forum: "ERDE: Entropy-Regularized Distillation for Early-exit"
_TMLR — Under review for TMLR_

### Review · Reviewer_cDaQ · 2026-04-26

**Summary Of Contributions:**

This paper proposes ERDE, a training framework that combines early-exit inference with knowledge distillation. The key idea is to modify the student objective based on whether the teacher prediction is correct. When the teacher is correct, the student is trained with the standard KL + CE objective. When the teacher is incorrect, the method replaces this objective at intermediate exits with an entropy-maximization term, encouraging the student to remain uncertain rather than imitate an incorrect teacher prediction. The intended effect is to reduce overconfident mistakes at shallow exits and route harder examples to deeper exits.

Experimentally, the paper shows that when $\theta = 0$, ERDE consistently outperforms both the student trained without KD and the student trained with standard KD. When $\theta > 0$, the method can substantially reduce computation relative to the teacher while maintaining relatively small accuracy degradation, and in some settings it achieves very large MAC reductions.

Strengths:
- The proposed entropy-based loss is well motivated. The method targets an important failure mode of vanilla KD in early-exit systems, namely that blindly matching the teacher can encourage overconfident intermediate predictions even when the teacher is wrong.
- The experimental results against the main internal baselines are promising. In particular, ERDE with $\theta = 0$ consistently improves over both no-KD and vanilla KD across the tested datasets, and in some cases even exceeds teacher accuracy.
- The method appears capable of achieving favorable accuracy-efficiency trade-offs in several settings.

Weaknesses:
- The empirical comparison against external baselines is limited. The paper mainly compares against vanilla KD variants and MSDNet, but does not sufficiently compare against other early-exit or dynamic inference methods.
- The ablation study does not fully isolate the contribution of the core design choices, especially the teacher-correct / teacher-incorrect branching rule and the removal of CE/KL on teacher-mistaken intermediate exits.

**Audience:**

Yes

**Audience Explanation:**

The paper addresses a relevant problem at the intersection of early-exit inference, knowledge distillation, and efficient model deployment. The idea of explicitly increasing uncertainty at intermediate exits when the teacher is wrong is conceptually interesting and should be relevant to readers working on efficient deep learning, dynamic inference, and model compression. Even though the current empirical support is not yet fully sufficient, the proposed perspective and preliminary findings would still be of interest to part of the TMLR audience.

**Broader Impact Concerns:**

I have no specific broader-impact concerns that would require an additional Broader Impact Statement.

**Claims And Evidence:**

No

**Claims Explanation:**

The claim that ERDE outperforms vanilla KD at $\theta = 0$ is reasonably well supported by Tables 1–3 across multiple datasets and architectures. This is the strongest part of the empirical evidence.

However, the paper also suggests that the entropy-based loss design is the main reason for the improvement, and this is not fully established by the current ablations. Figure 4 only studies sensitivity to $\omega_E$; it does not isolate the necessity of the teacher-correct / teacher-incorrect branching mechanism itself. For example, the paper does not show what happens if the branching rule is removed and entropy regularization is applied uniformly to all samples, or if CE/KL is removed without adding the entropy term. Without such controls, it is difficult to determine which part of the proposed recipe is actually responsible for the gains.

In addition, the paper implicitly suggests that ERDE is competitive with existing early-exit methods, but the empirical comparison is too narrow to support that broader claim. The experiments only compare against MSDNet and vanilla KD variants, without comparisons to other representative early-exit baselines such as BranchyNet or related dynamic inference methods.

Finally, the conclusion that $\omega_E = 5 \times 10^{-3}$ is the best choice is based on a single dataset and a single teacher-student pair. This is not enough to support it as a generally recommended hyperparameter.

**Requested Changes:**

-  Add stronger comparisons against external early-exit and dynamic inference baselines.  The current evaluation mainly compares ERDE against no-KD and vanilla KD variants under the same teacher-student setting, with only limited comparison to MSDNet. To support the broader empirical claims, the paper should include comparisons with more standard early-exit baselines and, if possible, representative dynamic inference methods.

- Provide more targeted ablations to isolate the contribution of the proposed loss design.
  The current ablations are not sufficient to validate the necessity of the main design choices in Eq. (4). In particular, the paper should isolate the effect of:  the teacher-correct / teacher-incorrect branching rule; removing CE/KL on teacher-mistaken intermediate exits.

- Clarify the method description in Section 3.
  The description of the early-exit threshold appears inconsistent with Algorithm 1. The text states that entropy is used as an uncertainty score and suggests that when the score falls below the threshold the sample continues through the network, which conflicts with the loop condition $c_i > \theta$. More generally, if entropy is indeed the uncertainty score, then decreasing the threshold should require greater confidence and therefore cause more samples to proceed to deeper exits, not fewer. This part of the method description should be carefully revised for correctness and clarity.

-  Expand the sensitivity analysis of ω_E and other hyperparameters.
   The current study of $\omega_E$ is limited to one dataset and one teacher-student pair. Evaluating whether the same trend holds across additional settings would make the conclusions more convincing.

---

### Review · Reviewer_WwPB · 2026-06-09

**Summary Of Contributions:**

The paper introduces ERDE (Entropy-Regularized Distillation for Early-exit), a method that trains a dynamic, early-exit student network using a larger early-exit teacher network. The key novelty lies in the training objective: for samples where the teacher model predicts incorrectly, an entropy-maximizing loss is applied to the student's intermediate exits to encourage uncertainty and prevent premature, incorrect exits. Experiments on CIFAR-10, CIFAR-100, and SVHN using ResNet and ConvNeXT architectures demonstrate that ERDE yields a better accuracy-efficiency (MACs/latency) trade-off compared to standard knowledge distillation.

**Additional Comments:**

The paper is well-written, clearly structured, and the notation in the methodology section is easy to follow. I lean to accept this paper based on current submission.

**Audience:**

Yes

**Audience Explanation:**

Model compression, dynamic neural networks, and knowledge distillation are highly active and relevant areas of research.

**Broader Impact Concerns:**

There are no significant ethical implications or broader impact concerns associated with this work.

**Claims And Evidence:**

Yes

**Claims Explanation:**

The authors provide extensive empirical evidence to support their claims. The ablation studies and baseline comparisons (teacher, student w/o KD, student w/ standard KD) are thorough. The results consistently demonstrate that the proposed entropy loss improves accuracy while reducing MACs and latency across different architectures (ResNet, ConvNeXT).

**Requested Changes:**

Large-Scale Dataset Evaluation: Relying solely on CIFAR-10/100 and SVHN is a significant limitation for a modern computer vision paper. The authors should evaluate ERDE on a larger-scale dataset, such as ImageNet, particularly since they are utilizing architectures like ConvNeXT that are designed for such scales.

Statistical Significance: The accuracy improvements over standard KD are sometimes marginal (e.g., ~0.5% to 1%). Including confidence intervals or error bars computed across multiple random seeds would heavily strengthen the empirical claims.

Deeper Analysis of MSDNet Comparison: Please provide a more detailed discussion on why ERDE struggles to consistently outperform MSDNet on CIFAR-10 (Figure 3a) compared to its stronger relative performance on CIFAR-100.

Hyperparameter Sensitivity: Briefly discuss how sensitive the threshold $\theta$ is to different datasets and whether it requires heavy tuning in practice.

---

### Review · Reviewer_G9DH · 2026-06-20

**Summary Of Contributions:**

This paper introduces ERDE (Entropy-Regularized Distillation for Early-exit), a model compression framework that combines Knowledge Distillation (KD) and Early-exit (EE) architectures. The primary contribution is a novel training loss that splits samples into two cases based on teacher correctness.

# Strength
(+) The core idea of using teacher error to regularize intermediate exit uncertainty ($\mathcal{L}_{E}^i$) is highly intuitive and addresses a practical pain point in early-exit training.

(+) The paper is well-structured and provides clear pseudo-code (Algorithm 1) alongside a comprehensive diagram (Figure 1) of the training workflow.

(+)  The empirical results demonstrate substantial computation trade-off efficiency, yielding up to a $4\times$ reduction in hardware latency.

# Weakness
(-) The baseline results, particularly on the CIFAR100 dataset, exhibit highly abnormal anomalies where larger teacher models severely underperform smaller student models.

(-) The choice of an unweighted sum for joint multi-exit training appears to heavily degrade the backbone networks, casting doubt on whether the reported improvements are true algorithmic gains or merely artifacts of suboptimal baselines.

**Additional Comments:**

Overall, I think the introduced method is interesting. I am looking for more solid results to support the author's motivation.

**Audience:**

Yes

**Audience Explanation:**

Researchers focusing on dynamic neural networks, efficient edge computing, and knowledge distillation will find the concept of entropy-regularized early exiting valuable.

**Broader Impact Concerns:**

No.

**Claims And Evidence:**

No

**Claims Explanation:**

Explain your answer above:
While the conceptual framework of ERDE is sound, the evidence presented in the experimental section is unconvincing due to significant discrepancies in baseline performance, especially in Table 2 (CIFAR100):

1. Performance Inversion (Teacher vs. Student): In the ConvNeXT setup on CIFAR100, the CNXT-b Teacher achieves an accuracy of only 0.7029. Strikingly, the CNXT-t Student without KD (completely independent training) outperforms its own teacher with an accuracy of 0.7146. In standard knowledge distillation literature, a student model trained from scratch should not naturally outperform a significantly larger, pre-trained teacher architecture by such a margin unless the teacher was fundamentally undertrained or misconfigured.

2. Model Scale Contradiction: Looking at the ResNet architectures in Table 2, the ResNet18 Teacher achieves 0.6758 accuracy, while the larger and more capable ResNet34 Teacher drops to 0.6654. This "the larger the model, the worse the accuracy" trend indicates a critical failure in the baseline training pipeline.

3. Suboptimal Absolute Baselines: A ConvNeXT-Base model (CNXT-b) starting with ImageNet pre-trained weights typically yields upwards of 80–85%+ accuracy on CIFAR100 under standard fine-tuning setups. The reported 70.29% strongly suggests a flawed evaluation or optimization environment.

The authors state on page 7 that they minimize the total loss as an unweighted sum of all exits. This unweighted formulation likely causes massive gradient interference from difficult, early exits to the deeper backbone layers, destroying the teacher's final performance. Because the baselines are severely degraded by this training choice, the claim that ERDE systematically outpaces standard training is not fully justified by sound evidence.

**Requested Changes:**

1. Sanity Check and Re-train Baselines: The authors can re-evaluate and re-train the teacher and baseline models to match community-standard accuracies on CIFAR100. If CNXT-b continues to score around 70%, a clear, thorough explanation must be provided as to why the pre-trained weights failed to yield standard downstream transfer results.

2. Resolve Performance Irregularities: Address the contradictions in Table 2 where ResNet34 underperforms ResNet18, and explain how a student without teacher guidance (w/o KD) successfully beats the teacher baseline.

3. Ablation on Multi-Exit Loss Weighting: Since the unweighted sum ($\sum \mathcal{L}^i$) appears to sabotage the backbone network, the authors should include an ablation study comparing their unweighted approach against a standard weighted multi-exit objective (e.g., assigning lower weights to earlier exits) to isolate whether ERDE's benefits hold up against properly optimized backbones.


4. Please provide standard deviations or error bars across multiple random seeds for Table 1, 2, and 3 to prove statistical significance.

5. Scale Evaluation: Testing the framework on a slightly more complex, modern dataset (such as Tiny-ImageNet or a subset of ImageNet-1K) would greatly strengthen the claim of "optimizing the trade-off between accuracy and efficiency" beyond low-resolution images.